# Validation and Spanish Adaptation of the Resilience Scale ER-23 in a University Population

**DOI:** 10.3390/healthcare13080886

**Published:** 2025-04-12

**Authors:** Isabel Ramírez-Uclés, Julia Otero, F. Pablo Holgado-Tello, Lucas Muñoz-López, María B. Sánchez-Barrera

**Affiliations:** 1Department of Personality, Assessment and Psychological Treatment, Universidad Nacional de Educación a Distancia (UNED), 28040 Madrid, Spain; 2Brain, Main and Behaviour Research Center, University of Granada, 18071 Granada, Spain; juliaotero@ugr.es (J.O.); lucasml@ugr.es (L.M.-L.); mbsanch@ugr.es (M.B.S.-B.); 3Department of Behavioral Sciences Methodology, Universidad Nacional de Educación a Distancia (UNED), 28040 Madrid, Spain; pfholgado@psi.uned.es; 4Department of Personality, Assessment and Psychological Treatment, University of Granada, 18071 Granada, Spain

**Keywords:** resilience, scale, validation, psychological well-being

## Abstract

**Background**: Resilience has received considerable attention in recent years and is a psychological characteristic that favors positive adaptation to adversity. **Objective**: The objective of this work is to generate a Spanish adaptation of the Resilience Scale (ER, acronym in Spanish) and to study the dimensionality of the scale through confirmatory factor analysis (CFA). **Methods**: The ER was administered to 1058 young Spanish people. The original English version of the Resilience Scale (25 items) was translated into Spanish, and the translation was confirmed through a backtranslation. The original version consists of two major factors: personal competence and acceptance of oneself and life. **Results**: The results confirm the goodness of the psychometric characteristics of the scale (internal consistency and criterion validity) with 23 items and the original two-factor model proposed by the authors. There was a positive correlation between the ER-23 adaptation and the Connor–Davidson Resilience Questionnaire (2002) and subjective psychological well-being and a negative correlation between the ER-23 adaptation and anxiety. **Conclusions**: The ER-23 adaptation is a valid and reliable tool that can be used in future research in a university population.

## 1. Introduction

Resilience has been defined as the ability of an individual to recover and adapt positively to stressful events or adversities [1,2]. It has been described as a dynamic and active process that involves adapting to or effectively managing important sources of stress or trauma [3,4,5,6].

Resilience is negatively associated with indicators of psychological distress, such as depression and anxiety [7,8,9,10,11,12,13], and positively associated with subjective well-being and positive emotions [14,15,16,17,18,19].

There are several tools adapted and validated in different languages, included Spanish, for measuring resilience in different contexts [20]; for example, the Brief Resilience Scale [21], the Connor–Davidson Resilience Questionnaire [22], the Adult Resilience Scale [23], the Brief Resilient Coping Scale [24], and the Resilience Scale [2].

The Wagnild and Young Resilience Scale [2] was the first evaluation tool developed and is one of the most used and accurate for measuring resilience [25]. The scale has two main factors: the first factor refers to personal competence (17 items) and includes self-confidence, independence, determination, invincibility, power, ingenuity, and perseverance and the second factor refers to the acceptance of oneself and of life and is composed of adaptability, flexibility, balance, and a stable perspective of life. The Cronbach’s α coefficient for the entire scale was 0.91 [2,26].

The Resilience Scale has been translated into different languages and adapted to different cultures, both in the original (RS25; see Table 1) and in the short version (RS14). Regarding the Spanish language, the scale has been validated on three occasions [27].

Throughout the years, various difficulties have been encountered in the evaluation of resilience [28,29,30]. Each new tool has been built taking into account previous instruments, with improvements in its characteristics; however, the Wagnild and Young Resilience Scale [2] presents unstable and unclear factorial structures in different languages [31]. In some studies, the two factors of the original scale have been confirmed [27,32,33,34], but in others, more than two factors have been found [32,35,36,37,38], and others have demonstrated a one-dimensional structure [39,40,41,42]. These inconsistent results can be explained by cultural factors that differentiate the populations studied. The appearance of different factors in investigations and the elimination of different items in validation studies could be explained by the translation, cultural factors, and the population studied.

The main objective of this study was to adapt the 25-item original Resilience Scale [2] to Spanish, validate the new version using a population of young students at the University of Granada, and study the dimensionality of this scale through confirmatory factor analysis (CFA).

The second objective was to determine the relationship between the Resilience Scale and anxiety and subjective psychological well-being. Resilience Scale scores were expected to correlate positively with Connor and Davidson’s [22] resilience questionnaire scores and with subjective psychological well-being and correlate negatively with anxiety.

## 2. Method

### 2.1. Participants

The sample consisted of 1058 students from the University of Granada between 18 and 30 years of age (M = 20.02 years, standard deviation (SD) = 2.14; 793 women and 265 men). The students were toward earning the following degrees: psychology (33.1%), speech therapy (16.97%), physical activity and sports sciences (8.5%), marketing (7.8%), pharmacy (3.7%), physical therapy (6%), criminology (6%), nursing (3.7%), labor relations (2.9%), and computer science (1.1%). Non-random convenience sampling was used to select the sample of young people. The CFA power analysis (G*Power) with 1058 participants confirmed a statistical power of 1.0, ensuring robust and reliable results.

The participants provided written informed consent, accepted the study conditions, and understood data confidentiality, thus complying with the data protection regulations. The Human Research Ethics Committee of the University of Granada approved the study. The target population of the study is young Spanish people (18–30 years) from Granada University. Once selected according to their resilience scores (high and low), a subsequent psychological study was carried out in laboratory sessions. Nine participants were excluded from the initial sample because they were over thirty years old.

### 2.2. Instruments

The Resilience Scale (ER-25; [2]) consists of 25 closed-ended items scored using a Likert scale, with seven choices ranging from 1 (strongly disagree) to 7 (strongly agree). Higher scores indicate greater resilience, with the total score ranging from 25 to 175: values greater than 147 indicate high resilience, values between 121 and 146 indicate moderate resilience, and values less than 121 indicate lower resilience. The 25 items are divided into two factors: personal competence (17 items) and acceptance of oneself and life (8 items). The first factor refers to trust, independence, determination, control, resourcefulness, and perseverance (items 1, 2, 3, 4, 5, 6, 9, 10, 13, 14, 15, 17, 18, 19, 20, 23, and 24), and the second factor refers to adaptability, balance, flexibility, and balanced life perspectives (items 7, 8, 11, 12, 16, 21, 22, and 25). Cronbach’s α coefficient was 0.91 for the scale. In our study, the mean scale score for the participants was 127 points. Percentile analyses have been carried out based on the resilience scores of the participants in the resilience questionnaire, with the cut-off point for determining low resilience being scores below 118 points and 138 for people with high resilience. Comparing these scores with those of the original scale would expound the differences by the age of the participants since our objective is to use a young sample, and therefore, their resilience scores are lower than those of adults [43].

The Connor–Davidson Resilience Scale (CD-RISC-25; [22]) consists of 25 items, each of which is rated on a five-point scale (0–4), ranging from not true (0) to almost all the time (4). Higher scores reflect greater resilience, with the total score ranging from 0 to 100. The questionnaire had good psychometric properties based on a validation study with a population in the United States (Cronbach’s α = 0.89).

The subjective Psychological Well-being Scale (PWBS; [44]) has a total of 39 items to which the participants respond using a response format with scores ranging from 1 (totally disagree) to 6 (totally agree), grouped into six categories: self-acceptance (recognition and acceptance of positive and negative traits), relationships (presence of close and stable relationships), autonomy (self-regulation of opinions and decision-making), control of the environment (management of daily responsibilities), personal growth (creation of conditions to develop his/her potential and evolve), and purpose in life (ability to clearly define objectives). The internal consistency was adequate, with Cronbach’s alpha = 0.83.

The State–Trait Anxiety Inventory (STAI-T; [45]) evaluates trait anxiety through 20 items answered using a Likert-type response scale with four choices. For state anxiety, the scale ranges from 0 (not at all) to 3 (a lot), while for trait anxiety, the scale ranges from 0 (almost never) to 3 (almost always). The items that are reverse scored evaluate well-being or the absence of anxiety, while the other items evaluate the presence of anxiety. The total score is obtained by adding the score for each item. The reliability (Cronbach’s alpha) was 0.90 for trait anxiety and 0.94 for state anxiety [46].

### 2.3. Procedure

The original 25-item Resilience Scale was translated into Spanish by two bilingual persons (one native-English speaker and one native-Spanish speaker). One individual translated the text from English to Spanish, and then the other translated the Spanish version back to English. With the help of two native-Spanish speakers, the translation was compared with the original English version of the questionnaire to detect grammar errors and errors that would affect the comprehension of the scale by a young population. The adaptation of the wording was carried out through an inter-judge agreement to preserve the original meaning of the item within the context of the study. The Spanish version was tested in a sample of students from the University of Granada to detect possible comprehension problems. Throughout the process, the guidelines of the International Testing Commission were followed for the adaptation of the questionnaire, and subsequently, the tool was administered to a final sample of participants from the University of Granada.

For this research, the need for honesty was emphasized to the participants before they completed the questionnaire, and the participants were provided with a guarantee that their answers were confidential and a reminder that their participation was completely voluntary. Regarding sample selection, all participants received information by this research about the experiment in a classroom as a group. All participants completed a demographic information sheet (age, gender, and education), the ER-25, CD-RISC-25, STAI-T, and PWBS in this respective order with a duration of approximately thirty-five minutes.

### 2.4. Data Analysis

To obtain empirical evidence regarding the construct validity of the questionnaire and given the ordinal nature of the data, confirmatory factor analysis (CFA) was performed using polychoric correlations and unweighted least squares (ULS) as estimation methods [47,48,49,50]. The basic psychometric properties of the dimensions obtained and of the items were also verified. For criterion validity, correlation analysis was performed to determine the relationship between the ER Scale and the dimensions and subdimensions of the CD-RISC 25, PWBS, and STAI-T.

The statistical programs SPSS 15.0 for Windows and LISREL 8.8 were used for the analyses [51].

## 3. Results

### 3.1. Descriptive Analysis of the Items

Table 2 provides the descriptive analysis of item scores. Most of the skewness and kurtosis indices indicate deviations from a normal curve, justifying the use of the robust unweighted least squares for the CFA.

### 3.2. Validity Tests Based on the Internal Structure

In accordance with the original structure proposed by Wagnild and Young [2], the dimensions of the instrument are grouped in personal competence and acceptance.

Based on the CFA results for the model, the following global goodness of fit indices were obtained: χ^2^ (*d.f.* = 274; *p* < 0.001) = 1546.67 (values should be non-significant or show a *p*-value > 0.05 for good fit); root mean square error of approximation (RMSEA) = 0.086 (90% C, 0.083 to 0.090) (values < 0.08 are adequate); goodness of fit index (GFI) = 0.95 (values > 0.90 are considered adequate); adjusted goodness of fit index (AGFI) = 0.94 (values > 0.90 are considered adequate); comparative fit index (CFI) = 0.72; normed fit index (NFI) = 0.70; non-normed fit index (NNFI) = 0.70 (values > 0.90 are adequate); and standardized root mean square residual (SRMR) = 0.07 (values < 0.10 are adequate).

These results, at the limit of what is considered a good fit, jointly with the problems that the two items have, suggest the need for a model re-specification, considering the elimination of both items. Item 17 had a standardized lambda of 0.09, while item 24 had a lambda of −0.31; in addition, the discrimination regarding dimensions were 0.08 and −0.22. The reliability index for item 17 was 0.11, and that for item 24 was −0.34. Finally, the validity index was 0.06 for item 17 and −0.25 for item 24. That is, from a measurement perspective, both items present serious problems, with both belonging to the same factor, i.e., personal competence. From a conceptual framework perspective, item 17, “Believing in myself helps me to overcome difficult moments”, refers to self-belief, and the other items that compose the factor refer to the measurement of personal competence. This is reflected in the wording of items as characteristics of the person rather than belief in oneself, for example, “I feel…”, “I am capable of…”, “I can…”, “I can…” In addition, item 24, i.e., “I have enough energy to do what I have to do”, could indicate a one-off state, but the factor is measuring self-confidence as a permanent trait (see Table 3).

Given the measurement and conceptual circumstances surrounding both items and given that the relevance and representativeness of the factor remain, both items were eliminated from the measurement model.

### 3.3. Confirmatory Factorial Analysis

To obtain empirical evidence regarding the adequacy of the postulated structure of the ER Scale, CFA was conducted after removing items 17 and 24. The dimensional structure considered implied a general second-order factor referring to resilience and two first-order factors. The following composed the two first-order factors: 15 items contributed to the first factor (personal competence) (1, 2, 3, 4, 5, 6, 9, 10, 13, 14, 15, 18, 19, 20, and 23), and 8items contributed to the second factor (acceptance) (7, 8, 11, 12, 16, 21, 22, and 25).

For the model examined, the global fit indices were as follows: χ^2^ = 2560.55; *d.f*. = 272; *p* = 0.00; RMSEA = 0.073 (90% CI, 0.070 to 0.076); GFI = 0.95; AGFI = 0.95; CFI = 0.93; NFI = 0.91; and NNFI = 0.92. Finally, the SRMR was 0.076. These data show that the fit values of the model are appropriate. All the lambda and gamma parameters were statistically significant (see Figure 1).

### 3.4. Psychometric Properties

Table 4 shows, using the model proposed in this work, the descriptive data and reliability of the scales obtained in the sample.

The global reliability indices for personal competence were as follows: McDonald’s ω = 0.82 and Cronbach’s α = 0.81. For acceptance, McDonald’s ω = 0.65 and Cronbach’s α = 0.63. Regarding the reliability of the total scale, McDonald’s ω = 0.86, and Cronbach’s α = 0.86.

In general terms, the values obtained for the items were good indicators of their psychometric properties. The reliability for factor 1 and the global scale were adequate; therefore, those scores could be used as a measure of global resilience and confidence. However, there was a measurement problem regarding the reliability of acceptance with item 8. First, the low reliability of this dimension may be due to underrepresentation; therefore, additional indicators of acceptance should be investigated. In this underrepresented dimension, item 8, although related to the dimension and, from a conceptual point of view, could be included in a certain way, could be irrelevant because its relation with the criteria used is very low. Therefore, the acceptance score should be interpreted with caution and in the context of other measurements.

### 3.5. Criterion Validity

To determine the criterion validity, we calculated the Pearson bivariate correlation indices for global resilience and the factors personal competence and acceptance and the CD-RISC-25, PWBS subscales, and STAI-T (see Table 4). There was a significant and positive correlation between the total and each subscale score of the ER-23 Scale and the variables resilience (CD-RISC 25), self-acceptance, environmental mastery, personal growth, purpose in life, and psychological well-being (PWBS). Furthermore, there was a negative correlation between the subscales positive relations, autonomy (PWBS), trait anxiety (STAI-T), and each of the ER-23 Scale factors.

Based on Rosnow and Rosenthal [52], the correlations with the Resilience Scale (CD-RISC-25) present a large effect size; however, given the large sample size, correlations of −0.07 or −0.08 (between autonomy (PWBS) and all factors in the ER Scale) are statistically significant.

The CFA results and the descriptive and psychometric indices of reliability and validity revealed that the structure of the ER-23 Scale supports the two-factor model of the original scale. These results are consistent with those from previous studies in different contexts [27,32,33,34,53].

The ER-23 Scale was positively correlated with the CD-RISC-25, potentially indicating that both can be useful to evaluate the resilience of young people or to researchers interested in determining the presence or absence of variables that may be involved in resilience. Likewise, both the total ER-23 Scale score, and factor scores were directly and significantly correlated with the subjective psychological well-being, a concept closely related to resilience [54,55], and with its dimensions of self-acceptance, control of the environment, personal growth, and life purpose. Within this scale of well-being, we highlight that the dimensions of positive relationships and autonomy are indirectly related to the resilience construct potentially because, in the Ryff model, the definitions of autonomy and positive relationships with others correspond to the basic needs of autonomy and relationships for any individual [56]. Taking this idea into account, Parra et al. [57] reported that autonomy is a key factor for a successful transition to adult life, consisting of behaviors (individual ability to act independently of others), cognitions (including self-efficacy, which empowers individuals to act in different areas of his life), and emotions (links built with others). Therefore, the acquisition of autonomy and the development of positive relationships with others play central roles in the psychological health and well-being of young people, and age modulates the intensity of this relationship [58,59], which may be one of the reasons why the association between resilience measured with both the ER Scale and the CD-RISC and these two dimensions of the PWBS were inversely correlated in the young population.

Resilience was also inversely associated with trait anxiety; this finding is consistent with research that has addressed this topic [7,41,60,61,62,63]. Recent evidence suggests that different psychological factors negatively link resilience with anxiety, promoting mental health and preventing the development of psychopathology despite being exposed to significant stressors [3,7,25,35,40,64,65,66,67].

The results of the present study provide evidence in favor of the validity of this measure of resilience and reveal that the version adapted to a young Spanish population replicates the original theoretical structure. Taking into account these results, in the future, the instrument designed herein will serve as a tool that is easy to apply and understand for young students. This adapted version generated satisfactory data, had a structure similar to the original theoretical model, and showed high internal consistency and validity. Given that resilience is associated with subjective well-being and negatively with psychological disorders [68,69,70,71,72,73], the use of this scale in young populations is considered relevant, mainly due to difficulties in the transition of young populations into adulthood.

Despite the evidence for reliability and validity for the ER-23, this study has several limitations. First, the sample used in this study was relatively large and homogeneous with respect to age and education level. The sample was not randomly selected, and even though it was large and homogeneous, it did not belong to the age and education level. The generalization of the results, therefore, is limited to a population of students; further validation is required in other adult samples. Although both men and women participated in this research, there is a high proportion of women, which could influence the measurements of resilience. For this reason, it is important to consider this variable to check for gender differences in the level of resilience and, therefore, generalize the result in other adult samples.

Regarding the implications of the study, the scale is applicable both in research and in psychological interviews, as well as in therapeutic work, where a rapid report of changes related to interventions performed is required. In future studies, it should be taken into account that the acceptance factor has a lower load of items; therefore, it is less represented than the personal competence factor, and more research should be conducted to identify additional indicators of acceptance. In addition, as previously mentioned, “I am my own friend” should be redefined. In this way, the interpretation of having one or more friends would be recorded.

Because resilience can change as a result of life experiences and interventions performed, longitudinal research is considered crucial, with the aim of obtaining more evidence of validity for this scale. In addition, the application of carefully analyzed and reviewed scales to samples with considerably different characteristics would be interesting.

We conclude that the version adapted to Spanish and validated in a sample of young people is a sensitive tool for evaluating levels of resilience. High ER-23 scores are associated with greater psychological well-being and lower levels of anxiety, which suggests that this adaptation can be recommended for use in young people.

## 4. Conclusions

The aim of this study was to develop and validate a Spanish version of the Wagnildand Young [2] Resilience Scale, adapted for use with a university population. The results demonstrate that the adapted version (ER-23) is a valid and reliable tool for measuring various aspects of resilience in this population, showing good applicability and understanding of the scale. The internal consistency of the ER-23 was excellent, with a Cronbach’s alpha coefficient of 0.86, aligning with findings from previous studies that have used this scale. Confirmatory factor analyses (CFAs) supported the two-factor structure, consistent with the original theoretical model of the scale. The results also indicated significant positive correlations between the ER-23 and subjective psychological well-being, as well as inverse correlations with trait anxiety, further supporting the theoretical association between resilience and various aspects of mental health. However, several limitations must be considered. Despite the strengths of this study, there are several important limitations that should be acknowledged. First, the external validity of the results is limited, as the sample used in this study was relatively large and homogeneous with respect to age and education level. The sample was not randomly selected, and even though it was large and homogeneous, it may not represent the broader population in terms of age and education level. Therefore, the generalization of the results is limited to a population of university students, and further validation is required in other adult samples. Additionally, while both men and women participated in this research, the sample had a high proportion of women compared to men. This imbalance could influence the measurements of resilience, as gender differences may affect the results. Therefore, it is important to consider this variable in future studies to assess gender differences in resilience and to ensure that the results can be generalized to other adult populations. Second, the study relied on self-reported data, and self-report biases such as Insufficient Effort Responding (IER) and Careless Responding (CR) were not directly addressed. These biases can distort the validity of the results. In future studies, we plan to implement strategies to detect and correct for these biases, including the use of additional validation measures and more rigorous monitoring of response quality. Third, this study does not address the possibility of conducting longitudinal research. Longitudinal studies would allow for a deeper understanding of how the scale performs over time. In future research, we will explore the feasibility of such an approach and will provide a more comprehensive discussion on the implementation of longitudinal designs to assess the scale’s stability and predictive power over time. Furthermore, the university context may have influenced the results, as university students may exhibit different resilience profiles compared to the broader population. This study did not specifically explore the potential impact of the university environment on resilience or its components. Future research should consider how the specific context of a university, including academic pressures and social dynamics, might shape resilience outcomes.

Despite these limitations, the adapted and validated ER-23 version for young Spanish students has proven to be a useful and sensitive tool for assessing resilience. Higher scores on the scale were associated with greater psychological well-being and lower levels of anxiety, suggesting its applicability in clinical and research contexts. This study highlights the need for longitudinal research to provide further evidence of the scale’s validity and its ability to capture changes in resilience over time, as well as its applicability in samples with more diverse demographic characteristics.

## Figures and Tables

**Figure 1 healthcare-13-00886-f001:**
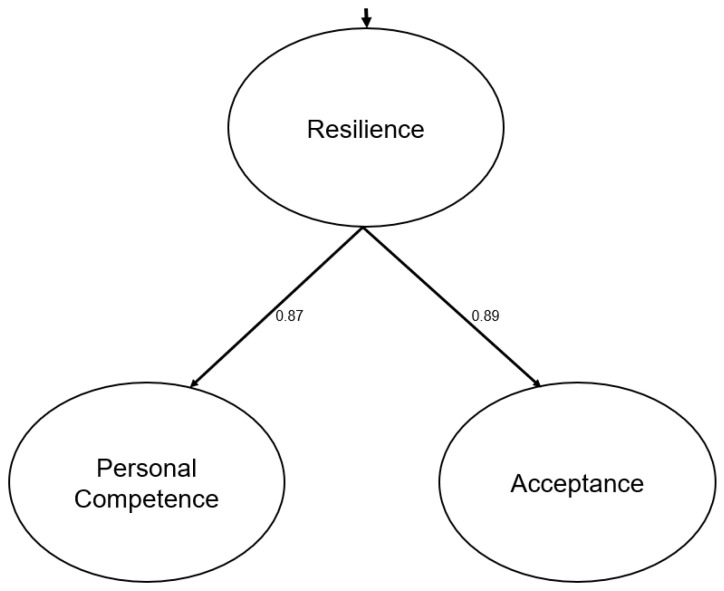
Completely standardized solution of the model.

**Table 1 healthcare-13-00886-t001:** (**a**) Descriptive statistics for item scores: Mean (M), Standard Deviation (SD), Skewness, and Kurtosis (Spanish); (**b**) Descriptive statistics for item scores: Mean (M), Standard Deviation (SD), Skewness, and Kurtosis (English).

(**a**)				
**Item**	**M**	**SD**	**Skewness**	**Kurtosis**
1. Cuando hago planes, los llevo a cabo	5.80	1.19	−1.46	3.26
2. Normalmente me las arreglo de una u otra forma	5.78	1.48	−1.31	1.13
3. Soy capaz de valerme por mí mismo, más que cualquier otra persona	6.27	1.10	−1.87	4.15
4. Me es importante mantener el interés por las cosas	4.62	1.60	−0.33	−0.62
5. Puedo estar solo si es necesario	5.11	1.66	−0.77	−0.18
6. Me siento orgulloso de haber logrado cosas en mi vida	5.10	1.36	−0.56	0.07
7. Generalmente me tomo las cosas con calma	4.94	1.47	−0.59	−0.13
8. Soy mi propio amigo	2.99	1.70	0.72	−0.40
9. Me siento capaz de hacer muchas cosas a la vez	5.13	1.29	−0.58	0.19
10. Soy resolutivo/decidido	5.37	1.36	−0.83	0.44
11. Rara vez me pregunto por el sentido de las cosas	5.32	1.40	−0.95	0.57
12. Afronto las cosas día a día	5.31	1.14	−0.91	1.24
13. Puedo superar las dificultades porque he pasado por experiencias difíciles en otras ocasiones	5.93	1.12	−1.26	2.33
14. Tengo autocontrol	5.55	1.40	−1.12	1.08
15. Mantengo el interés por las cosas	5.48	0.85	−2.50	9.88
16. Generalmente puedo encontrar algo de lo que reírme	5.73	1.23	−1.05	1.14
17. Creer en mí mismo me ayuda a superar los momentos difíciles (eliminado)	5.19	1.42	−0.96	0.72
18. Generalmente, soy una persona en la que la gente puede confiar en caso de emergencia	5.83	1.35	−1.35	1.73
19. Normalmente puedo ver una situación desde diferentes puntos de vista	3.37	1.74	0.47	−0.74
20. A veces me obligo a hacer cosas tanto si quiero como si no	4.77	1.17	−0.43	0.26
21. Mi vida tiene sentido	5.25	1.22	−0.63	0.33
22. No me preocupo por cosas sobre las que no puedo hacer nada	5.33	1.69	−0.82	−0.35
23. En general, cuando estoy en una situación difícil, sé cómo salir de ella	4.55	1.16	−0.84	0.59
24. Tengo energía suficiente para hacer lo que tengo que hacer (eliminado)	2.51	1.55	0.76	−0.62
25. No importa si hay personas a las que no le caigo bien	4.45	1.30	−0.60	1.40
(**b**)				
**Item**	**M**	**SD**	**Skewness**	**Kurtosis**
1. When I make plans, I carry them out	5.80	1.19	−1.46	3.26
2. I usually manage one way or another	5.78	1.48	−1.31	1.13
3. I am more capable of taking care of myself than anyone else	6.27	1.10	−1.87	4.15
4. It is important for me to maintain interest in things	4.62	1.60	−0.33	−0.62
5. I can be alone if necessary	5.11	1.66	−0.77	−0.18
6. I am proud of the things I have accomplished in my life	5.10	1.36	−0.56	0.07
7. I generally take things calmly	4.94	1.47	−0.59	−0.13
8. I am my own friend	2.99	1.70	0.72	−0.40
9. I feel capable of doing many things at once	5.13	1.29	−0.58	0.19
10. I am resolute/decisive	5.37	1.36	−0.83	0.44
11. I rarely question the meaning of things	5.32	1.40	−0.95	0.57
12. I face things day by day	5.31	1.14	−0.91	1.24
13. I can overcome difficulties because I have gone through tough experiences before	5.93	1.12	−1.26	2.33
14. I have self-control	5.55	1.40	−1.12	1.08
15. I maintain interest in things	5.48	0.85	−2.50	9.88
16. I can generally find something to laugh about	5.73	1.23	−1.05	1.14
17. Believing in myself helps me overcome difficult moments (removed)	5.19	1.42	−0.96	0.72
18. Generally, I am someone people can trust in case of an emergency	5.83	1.35	−1.35	1.73
19. I can normally see a situation from different points of view	3.37	1.74	0.47	−0.74
20. Sometimes I force myself to do things whether I want to or not	4.77	1.17	−0.43	0.26
21. My life has meaning	5.25	1.22	−0.63	0.33
22. I don’t worry about things I can’t do anything about	5.33	1.69	−0.82	−0.35
23. In general, when I’m in a tough situation, I know how to get out of it	4.55	1.16	−0.84	0.59
24. I have enough energy to do what I need to do (removed)	2.51	1.55	0.76	−0.62
25. I don’t mind if there are people who don’t like me	4.45	1.30	−0.60	1.40

**Table 2 healthcare-13-00886-t002:** (**a**) Standardized CFA results (Spanish); (**b**) Standardized CFA results (English).

(**a**)		
	**Resilience**	
**Item**	**F1**	**F2**
1. Cuando hago planes, los llevo a cabo	0.58	
2. Normalmente me las arreglo de una u otra forma	0.22	
3. Soy capaz de valerme por mí mismo, más que cualquier otra persona	0.61	
4. Me es importante mantener el interés por las cosas	0.33	
5. Puedo estar solo si es necesario	0.54	
6. Me siento orgulloso de haber logrado cosas en mi vida	0.59	
9. Me siento capaz de hacer muchas cosas a la vez	0.60	
10. Soy resolutivo/decidido	0.44	
13. Puedo superar las dificultades porque he pasado por experiencias difíciles en otras ocasiones	0.50	
14. Tengo autocontrol	0.71	
15. Mantengo el interés por las cosas	0.42	
18. Generalmente, soy una persona en la que la gente puede confiar en caso de emergencia	0.63	
19. Normalmente puedo ver una situación desde diferentes puntos de vista	0.22	
20. A veces me obligo a hacer cosas tanto si quiero como si no	0.68	
23. En general, cuando estoy en una situación difícil, sé cómo salir de ella	0.55	
7. Generalmente me tomo las cosas con calma		0.61
8. Soy mi propio amigo		0.05
11. Rara vez me pregunto por el sentido de las cosas		0.50
12. Afronto las cosas día a día		0.56
16. Generalmente puedo encontrar algo de lo que reirme		0.37
21. Mi vida tiene sentido		0.81
22. No me preocupo por cosas sobre las que no puedo hacer nada		0.27
25. No importa si hay personas a las que no le caigo bien		0.37
(**b**)		
	**Resilience**	
**Item**	**F1**	**F2**
1. When I make plans, I carry them out	0.58	
2. I usually manage one way or another	0.22	
3. I am more capable of taking care of myself than anyone else	0.61	
4. It is important for me to maintain interest in things	0.33	
5. I can be alone if necessary	0.54	
6. I am proud of the things I have accomplished in my life	0.59	
9. I feel capable of doing many things at once	0.60	
10. I am resolute/decisive	0.44	
13. I can overcome difficulties because I have gone through tough experiences before	0.50	
14. I have self-control	0.71	
15. I maintain interest in things	0.42	
18. Generally, I am someone people can trust in case of an emergency	0.63	
19. I can normally see a situation from different points of view	0.22	
20. Sometimes I force myself to do things whether I want to or not	0.68	
23. In general, when I’m in a tough situation, I know how to get out of it	0.55	
7. I generally take things calmly		0.61
8. I am my own friend		0.05
11. I rarely question the meaning of things		0.50
12. I face things day by day		0.56
16. I can generally find something to laugh about		0.37
21. My life has meaning		0.81
22. I don’t worry about things I can’t do anything about		0.32
25. I don’t mind if there are people who don’t like me		0.37
**Personal Competence**	**0.87 ***	
**Acceptance**	**0.89 ***	

* Factor loadings of the Personal Competence and Acceptance factors within the confirmatory factor analysis (CFA).

**Table 3 healthcare-13-00886-t003:** Items analysis.

Items	SD	Item-Rest Correlation	McDonald’s ω	Cronbach’s α	Reliability Index	Validity Index
Personal Competence						
ER1	1.14	0.3	0.78	0.76	0.34	0.31 **
ER2	1.42	0.2	0.79	0.77	0.28	0.13 **
ER3	1.04	0.49	0.76	0.74	0.51	0.49 **
ER4	1.53	0.32	0.78	0.76	0.49	0.24 **
ER5	1.6	0.5	0.76	0.74	0.8	0.37 **
ER6	1.35	0.47	0.76	0.74	0.63	0.50 **
ER9	1.25	0.49	0.76	0.74	0.61	0.46 **
ER10	1.36	0.35	0.77	0.75	0.47	0.42 **
ER13	1.08	0.4	0.77	0.75	0.43	0.33 **
ER14	1.36	0.58	0.75	0.74	0.79	0.57 **
ER15	0.8	0.28	0.78	0.76	0.22	0.27 **
ER18	1.31	0.48	0.76	0.74	0.63	0.52 **
ER19	1.71	0.21	0.79	0.77	0.36	0.17 **
ER20	1.19	0.6	0.76	0.74	0.71	0.58 **
ER23	1.14	0.45	0.76	0.75	0.51	0.47 **
**Scale ***	**0.78**	**---**	**0.82**	**0.81**	**---**	**---**
**Items**	**Acceptance**					
ER7	1.5	0.46	0.46	0.55	0.69	0.56 **
ER8	1.68	0.03	0.03	0.68	0.05	−0.02
ER11	1.32	0.4	0.4	0.57	0.53	0.38 **
ER12	1.16	0.38	0.38	0.58	0.44	0.44 **
ER16	1.2	0.19	0.19	0.63	0.23	0.28 **
ER21	1.22	0.52	0.52	0.54	0.63	0.58 **
ER22	1.65	0.29	0.29	0.61	0.48	0.26 **
ER25	1.28	0.31	0.31	0.6	0.4	0.34 **
**Scale ***	**0.89**	**---**	**0.65**	**0.63**	**---**	**---**
**Total Scale ***	**0.95**	**---**	**0.86**	**0.86**	**---**	**---**

* Values for the subscales and for the total scale; ** *p* < 0.01 (two-tailed).

**Table 4 healthcare-13-00886-t004:** Pearson’s correlations between personal competence, acceptance, and total resilience scores and scale and subscale scores for the CD-RISC 25, PWBS, and STAI-T.

	Personal Competence	Acceptance	Resilience
**CD-RISC 25**			
Resilience	0.73 **	0.65 **	0.75 **
**PWBS**			
Self-acceptance	0.13 **	0.11 **	0.14 **
Positive relations	−0.18 **	−0.21 **	−0.20 **
Autonomy	−0.08 **	−0.07 *	−0.08 **
Environmental mastery	0.16 *	0.17 **	0.17 **
Personal growth	0.39 **	0.35 *	0.40 **
Purpose in life	0.10 **	0.13 **	0.17 **
Psychological well-being	0.23 **	0.21 **	0.24 **
**STAI-T**			
Trait anxiety	−0.26 **	−0.21 **	−0.26 **

** *p*< 0.01 (two-tailed), * *p*< 0.05 (two-tailed). Note. CD-RISC 25 = Connor–Davidson Resilience Scale; PWBS = Psychological Well-being Scale; STAI-T = State–Trait Anxiety Inventory.

## Data Availability

The data or result files can be requested from the authors.

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
