# Peer review of "Validation and Spanish Adaptation of the Resilience Scale ER-23 in a University Population"

_healthcare, 2025, doi:10.3390/healthcare13080886_

Round 1

Reviewer 1 Report

Comments and Suggestions for Authors

PLEASE SEE ATTACHED REVIEW REPORT.

Author Response

Dear Rewiewer,

We appreciate the thorough and comprehensive review you have conducted of our research. In the attached document, we answer the questions raised.

Reviewer 2 Report

Comments and Suggestions for Authors

This study examined the reliability and validity of the Spanish version of the ER-23 in young people. Overall, this is a well-conducted validation study. The results of the CFA indicate that the ER-23 is a valid measurement tool for assessing resilience. Below are my comments:

  1. In the Introduction, the authors indicate that the ER scale has been validated in a Spanish population. However, what were the findings of previous studies? It is necessary to clarify how the present validation study aims to improve upon prior research conducted in Spanish-speaking populations.

  2. Lines 81–82: One point of confusion is that CFA is not intended to "study" dimensionality but rather to "confirm" a factor structure in a specific dataset— in this study, the original two-factor model. Given the background discussion on inconsistencies in factor structure of the ER-25, the current study objective may lead readers to question whether an EFA would have been a more appropriate approach.

  3. The data analysis section should specify which model fit indices were calculated and their cutoff values (e.g., RMSEA < 0.08, CFI > 0.90, etc.).

  4. Tables 1 and 2: While the study employs a Spanish version of the ER-23, providing English translations of each item would enhance comprehension for international readers.

  5. Table 2: The meaning of the values 0.87 and 0.89 should be clarified within the table.

  6. The rationale for removing the two items is well explained.

  7. **Please include a diagram of the final CFA model for readers.**

  8. The Discussion section is well written, effectively summarizing the results and discussing their implications for future research, as well as the study’s limitations. Well done!

Author Response

Dear Rewiewer,

We appreciate the thorough and comprehensive review you have conducted of our research. In the attached document, we answer the questions raised

Round 2

Reviewer 1 Report

Comments and Suggestions for Authors

Please see attached report.

Author Response

Dear Reviewer,
We appreciate your thorough and detailed review. In the attached file, we have addressed the suggestions you made in your second review.
